# Burdens and Resources of Staff of a Specialized Ward for Neuropalliative Care: A Cross-Sectional Survey

**DOI:** 10.3390/brainsci12121697

**Published:** 2022-12-10

**Authors:** Sarah Herwest, Stella Linnea Kuhlmann, Anna-Christin Willert, Christoph Johannes Ploner, Alexander Bernhard Kowski

**Affiliations:** Department of Neurology, Charité-Universitätsmedizin Berlin, 13353 Berlin, Germany

**Keywords:** palliative care, neurology, care giver burden, resilience, resources, communication

## Abstract

Palliative care adds significant burdens to healthcare workers. In neuropalliative care (NPC), additional challenges include patient symptom burdens, such as impairments in mobility, cognition, and communication. After one year of operating the first NPC ward in Germany, we assessed burdens, resources, and the number of deaths perceived as stressful. NPC physicians and nursing staff were compared with the team of other neurological wards, including a stroke unit. The assessment took place between March 2022 and May 2022. All 64 team members were contacted; the responder rate was 81%. The perceived burden was high but did not differ between groups. There were no differences between the NPC- and the neurological wards in the number of deaths perceived as stressful. However, rather than the number of deaths, the circumstances of dying influence the perceived distress. The resources mentioned were similar between groups, with the team and private life being most important. Communication difficulties were frequently cited as stressful, whereas successful communication was identified as a resource.

## 1. Introduction

Palliative care becomes increasingly relevant as the number of patients suffering from advanced incurable diseases increases due to demographic change, longer life expectancy, and improved medical care [1]. This also includes the young discipline of neuropalliative care (NPC) [2]. To assure high-standard care for the increasing number of patients with palliative care needs, a sufficient number of motivated and resilient staff is needed. However, the overall burden on healthcare professionals is perceived as high [3,4]. With the emergence of palliative care, contributing factors of additional strain have been postulated. These comprise the day-to-day handling of patients with serious illnesses and complex symptom burdens, ethically demanding decisions, a high level of expertise, as well as interpersonal skills [1]. In NPC, there are additional demands on healthcare professionals due to the patient’s impairments in communication, mobility, behavior, and challenges in prognostic assessment in comparison to palliative care for cancer patients [2]. A high sense of workload, stress, and dissatisfaction at work lead to overall dissatisfaction with life, poorer health, and limited capacity for teamwork but ultimately to poorer patient care and more healthcare workers changing their employment [3,5,6,7]. In studies examining burnout, emotional exhaustion, stress, and depression in palliative care, high levels of the examined symptoms have been found [8,9,10,11]. Interestingly, compared to other disciplines, similar [3,4,9] or even lower levels [1,3] of burnout symptoms and subjective stress have been found in palliative care professionals. This might point to increased resources of palliative care teams. Care for dying patients and their families can be a source of job satisfaction, and thus working in palliative care is often perceived as particularly meaningful [3].

In Germany, burdens and protective factors of palliative care teams were studied in 2009 by Müller and colleagues, and in 2020 by Ateş and colleagues [12,13]. In 2009, a nationwide survey was conducted in hospices and palliative care units. This survey included questions on stress, including the number of deaths perceived as stressful, protective factors, and prospects of the staff [12]. In the 2020 survey, hospices, palliative care units, and outpatient palliative care structures were included. Through validated questionnaires adapted to palliative care and after an expert workshop, the questionnaire used in 2009 was updated and included additional stress factors, indices of workload, questions on personal well-being, the subjective development of working conditions, and an assessment of personal attitude toward palliative care [13].

However, these factors have not been investigated in NPC teams so far. Here, we investigated whether pooling of neurologically severely ill patients in a NPC ward puts particular demands on healthcare workers in comparison to teams on other neurological wards, including a stroke unit. We assessed the relevance of and responses to stress factors, including the number of deaths per week, and resources for day-to-day work. 

## 2. Materials and Methods

We surveyed physicians and nursing staff on the NPC ward and the other neurological wards, including the stroke unit (neuro ward) of the Department of Neurology, Campus Virchow-Klinikum in Berlin (Germany), between March 2022 and May 2022. Approval was given by the human resources department (Charité–Universitätsmedizin Berlin). A slightly adjusted version of the questionnaire by Ateş et al. [13] was used. It consisted of 95 questions, each with an option for comments. Items targeted the critical number of deaths per week, potential burdens, personal resources, institutional and interpersonal burdening events, stress symptoms after being confronted with death, palliative care attitude, change of working conditions, team sustainability, and physical and mental distress symptoms. Responses were given on 4- to 5-point Likert scales. Sociodemographic and structural information was requested. Free comments on the survey were optional. The following adjustments to the original questionnaire by Ateş et al., were made: It was digitalized (SoSci Survey GmbH, Munich, Germany), and the wording was adapted to our inpatient context and main addressees (physicians and nurses). Questions on demographic data were reduced to ensure anonymization; a question specifying the respondents’ workspace was added. The exact changes made to the questionnaire can be found in Appendix A. The questionnaire was sent to the business mail addresses of the addressed physicians and nursing staff as an anonymized link.Regarding the response rate needed for a valid team analysis, we applied the required response rate of at least 75–80% for “small companies” (corresponding to 10–49 employees per ward) [13]. Descriptive statistics were performed using IBM SPSS Statistics Version 29.0 (International Business Machines Corporation, Armonk, NY, USA). Means are given with standard deviation (SD). For group comparison, 4-point Likert scale items were dichotomized, and 5-point Likert scale items were reduced to three tendencies. To compare the main tendencies between the NPC and neuro wards, all questioned variables of the respective category (burdens, personal resources…) were pooled and compared as described. For group comparisons, the Mann–Whitney U test was used for continuous variables after testing for normal distribution (Shapiro–Wilk test). Fisher’s exact test was used for categorical variables. In addition, *p*-values less than 0.050 were considered statistically significant. Where necessary, significance levels were corrected for multiple testing using Bonferroni correction.

## 3. Results

### 3.1. Sample Description

Overall, we contacted 64 persons, 23 working on the ward for neuropalliative care (NPC ward), and 41 working on other neurological wards, including the stroke unit (neuro ward). Fifty-two (=81%) persons completed the questionnaire. The response rates were sufficient for valid team analysis, with 87% on the NPC ward and 78% on the neuro ward. For a detailed sample description, see Table 1. Mean work experience differed (not statistically significant) between the two wards (NPC ward 17.7 years, neuro ward 11.1 years, *p* = 0.1). However, the mean experience in palliative care was significantly higher on the NPC ward (NPC ward 4.7 years, neuro ward 0.9 years, *p* = 0.019). Most survey participants worked full-time. Respondents worked 5.5 h of overtime per week, with no difference between groups (*p* = 0.8). About one-third of the respondents of both groups lived alone, and 20-30% were involved in the care of relatives. A statement about the age or sex distribution cannot be made because anonymization precluded the evaluation of these demographic variables.

### 3.2. Critical Number of Deaths per Week

The mean number of deaths per week that was personally endurable without becoming overburdened did not differ (NPC ward: 3, SD 2.35; neuro ward: 4, SD = 2.43; *p* = 0.106). However, nearly one-fourth of all team members noted that rather than the number of deaths, the circumstances of dying were of greater importance for stress prevention. One team member stated: “*Accompanying dying patients, provided that the therapy concept had been discussed well, is an honor that fills me with pride.*”

### 3.3. Potential Burdens

We measured potential burdening factors on an organizational, personal, collegial, and patient- and relative-specific level (Appendix A). Overall, the perceived burden on the NPC team was not higher compared to the staff of the neuro ward (*p* = 0.07). The most burdening factors on both wards differed only slightly. The factors perceived as “much” or “very much burdening” by at least one-third of both teams are shown in Figure 1. These included organizational factors (“*Too little staff*”, “*High documentation effort*”), excessive demands on one’s own work (“*Too high demands of others on my care*”, “*Too high own demands on my care*”), therapy-associated factors (“*Conflicting treatment plans within the care network*”, “*Unsuccessful treatment of symptoms*”, “*Start or continuation of therapies in patients with poor prognosis*”), lack of time in various contexts (“*Too little time for rituals*”, “*No time to address wishes and problems of those affected*”), as well as social/communication factors “*Particularly difficult relationships with patients and relatives*”, “*Communication difficulties*”, and “*Conflicts in the (affected) families*”.

Regarding the significant differences in perceived burden, the NPC team was less burdened with “*Too many patients”* (*p* < 0.0001, significance level after Bonferroni correction *p* < 0.0016), “*No time to address wishes and problems of those affected*” (*p* = 0.05, after Bonferroni correction n.s.), “*Socially isolated patients*” (*p* = 0.042, after Bonferroni correction n.s.), “*Low/lack of family support*” (*p* = 0.010, after Bonferroni correction n.s.), and “*Patients with good prognosis rejecting suggested therapies*” (*p* = 0.05, after Bonferroni correction n.s.). Neither team felt (very) much burdened with “*Too few patients*”, less than one-fifth with “*Feelings of guilt*” and “*Accompanying deceasing patients for too short a time*”, and less than one-fourth with an “*Accumulation of deaths*” and “*Memories of deceased persons in their own environment*”.

Only a few respondents took the opportunity to specify diseases that they found difficult to care for. Of these, 13 (NPC ward *n* = 8, neuro ward *n* = 5) considered dealing with patients with amyotrophic lateral sclerosis (ALS) to be personally burdening, and, nearly similarly frequently, tumor diseases (NPC ward *n* = 6, neuro ward *n* = 4). Less frequently (*n* = 3 in each group), care for patients with dementia was considered personally distressing.

### 3.4. Institutional and Interpersonal Burdening Events

Overall, there was a high “*Sense of purpose*” at work among the respondents, and more than half found it “*Satisfying*”. Half of the staff complained of “*Unmanageable workload*” *on both wards*. Experiencing “*Excessive responsibility*” often/always tended to be reported less often on the NPC ward than on the neuro ward (15% vs. 22%, *p* = 0.05). On the NPC ward, nearly three-quarters stated that they rarely or never had to take on too much responsibility, while on the neuro ward, over two-fifths had to do so sometimes, and nearly a quarter often or always. In contrast, around half of the respondents of each group sometimes “*Wished for more responsibility*”. “*Interprofessional work meets expectations*” often or always in 60% of the respondents of the NPC ward, and only in 37% of the neuro ward. On the NPC ward, nearly two-thirds of the respondents never or rarely “*Failed to partake in educational opportunities due to lack of resources*,” while on the neuro ward, nearly one-third failed to do so often, and one-quarter each sometimes or rarely. For detailed results, see Table 2.

### 3.5. Stress Symptoms Shown by the Team when Confronted with Death

We measured symptoms of stress possibly shown/exhibited when faced with death (Appendix A). Overall, there was no difference between the groups (*p* = 0.091). In both groups, the most frequent symptoms were “*Work-to-rule*”. “*Excessive talkativeness*”, “*Irritability*”, and “*Accusations*”. The overall least frequently shown symptoms in both wards were “*Crying*”, “*Refusal of new admissions*”, and “*Lack of involvement with other or new patients*”. Regarding the differences between both groups, “*Increased interprofessional tensions*” occurred less often on the NPC ward (NPC ward 5%, neuro ward 32%, *p* = 0.023 not significant after adjustment using Bonferroni correction).

### 3.6. Personal Resources

There were no significant differences in individual resources across both groups (*p* = 0.55); for detailed results, see Table 3. All resources were (very) important to nearly half of the respondents. The most important resources across both groups were the “*Team*”; “*Humor*”; several aspects of “*Private life*”, such as “*Family*”, and “*Meeting friends*”; and “*Experiencing nature*”. “*Supervision*”, “*Offers for self-care*”, “*Worldview/faith*”, “*Empathy of others*”, and “*Team activities”* were rated as (very) important in around half of the cases. “*Rituals*” and the “*Team*” seemed to be non-significantly more important on the NPC ward, whereas “*Sports*” was non-significantly more important on the neuro ward.

### 3.7. Palliative Care Attitude, Change of Working Conditions, and Team Sustainability

Over half of the respondents on the NPC ward and two-thirds on the neuro wardstated that their palliative care attitude had not changed since working in the current team (Table 4A). However, about one-quarter of the respondents indicated a change in attitude, specified as follows: Five respondents of the neuro ward mentioned an improvement in expertise and awareness (e.g., professionalism, dealing with death and therapy limitations), while five of the NPC ward indicated an improvement in soft skills (e.g., empathy and communication skills). Furthermore, two respondents of the NPC ward rated interprofessional collaboration as (much) improved, while two of the neuro ward perceived it as (much) worsened in the current team.

When asked about the consistency of the respective team in its current composition, there were no significant differences between groups (Table 4B). Most staff expected the team to last a few to many years. However, one team member of the neuro ward expected the team to last a few weeks, and over one-quarter expected the team to last a few months, whereas on the NPC ward, fewer than one-sixth thought so. One participant of the NPC ward criticized the wording of the question since changes in the team can also be invigorating as a team is not a permanent institution.

When asked about a subjective change in working conditions, the NPC team reported (strongly) improved conditions significantly more often than the neuro team (*p* = 0.033). In the neuro team, over half of the respondents perceived the conditions as unchanged (Table 4C).

### 3.8. Physical and Mental Distress Symptoms

There were no significant differences in physical and mental distress symptoms within the last four weeks between the NPC ward and the neuro ward (Appendix A). Overall, all symptoms of distress occurred (very) often in more than one-third of the participants. Over half of them felt “*Rushed respectively under time pressure*” and “*Completely exhausted after work*” (very) often (NPC ward 50/60%, neuro ward 69/57%). Overall, 40% (NPC ward) and 44% (neuro ward), respectively, reported that “*Sleep disturbances*” occurred (very) often. In total, 35% of the respondents on both wards felt “*Depressed*” and had “*Physical pain*” (very) often. However, around one-third felt “*Calm and balanced*” and had “*Lots of energy*” (very) often (NPC ward 35/30%, neuro ward 34/37%).

## 4. Discussion

In Germany, burdens and protective factors of palliative care teams were studied in 2009 by Müller and colleagues and in 2020 by Ateş and colleagues [12,13]. So far, these factors have not been investigated in neuropalliative care (NPC), i.e., a particularly demanding subdiscipline of palliative care [2]. This cross-sectional survey analyzes the burdens on the first neuropalliative ward in Germany and compares them to a neurological ward including, a stroke unit.

In 2009, the original idea behind the survey of palliative care units in Germany was the question of how much death a team can tolerate [12]. The mean number of personally endurable deaths was around 3–4 per week in our sample and the inpatient sample (palliative care and hospices) surveyed by Ateş and Müller and colleagues [12,13]. However, in the current study and the study by Ateş et al., many participants stated that a number of deaths can not be given because only the circumstances of dying are decisive in preventing stress [13]. Thus, the actual question to be asked appears not to be how much death a team can tolerate but under which circumstances a team can tolerate death.

The “*Sense of purpose at work*” on the NPC ward was slightly less pronounced than on the palliative care wards in Ateş et al., and the teams on both wards were less “*Satisfied with their jobs*” than the teams on the palliative care wards surveyed by Ateş and colleagues (95–98%) [13]. Further, around one-third often or always experienced an “*unmanageable workload*”. To improve satisfaction and the subjective workload, more control and self-determination of the employees of the job-demand-control model could be helpful. How those aspects can be implemented in an inpatient context remains to be discussed. “*Not being able to take part in educational offers due to lack of resources*” did not appear to be a major problem on either ward. Despite the high relevance given to interprofessional work in palliative care [14], “*Satisfaction with interprofessional work*” did not differ significantly between the two wards. However, “*Excessive responsibility*” occurred less frequently on the NPC ward and on the palliative care wards in Ateş et al. [13].

The staff of the NPC ward was not more burdened than the staff of the neuro ward, which is consistent with the existing literature. However, similar to the literature, our survey also showed a high overall sense of burden [1,3,4,8,9,10,11]. The most relevant stress factors were relatively consistent with the data of the palliative care wards in Ateş et al. [13]. “*Communication difficulties*” should be specified through qualitative studies, communication skills ought to be improved through in-service training, and regular multidisciplinary meetings must be implemented [15]. Further, handling “*Difficult relationships with patients and their relatives*” should obtain a more prominent role in the education of healthcare professionals. Regarding the “*Documentation effort*”, the relevance of the documentation needs to be made clear, and double documentation should be eliminated. To make work more efficient and free up more “*Time*”, inefficient processes should be identified and optimized, and digital solutions should be implemented where possible. Undergoing the additional palliative care qualification might reduce therapy-associated burdening factors on a (neuro)palliative care ward. “*Demands on one’s work*” should be regularly made aware of and compared with reality. Diverging demands should be communicated and worked on, as excessive demands are known stressors that can lead to a gratification crisis (effort-reward-imbalance model), less job satisfaction, and ultimately poorer health [16]. Consistent with Ateş et al., amyotrophic lateral sclerosis (ALS) and cancer were named as especially burdening diseases to accompany [13]. Qualitative interviews should investigate why accompanying patients with these diseases are perceived as burdening. The burden from “*Too many patients*” and “*No time to address wishes and problems of those affected*” was lower on the NPC ward not only compared to the neuro ward but also the palliative care wards in Ateş et al. [13]. Since the NPC ward was operated for only one year and with a staffing level corresponding to the staffing ratio, the difference might be artificial and should be validated in another survey after a longer period of operation. The staff of the NPC ward was less burdened with “*Socially isolated patients*” and “*Low/lack of family support*”. In Ateş et al., the staff of the palliative care wards was also similarly less burdened with that than the staff of the other services. This difference and its cause should be further investigated.

With few exceptions, the team reactions to the burden of deaths were similar across both wards, especially the most common ones. “*Excessive talkativeness*” and “*Irritability*” were among the most frequently occurring symptoms in the 2009 and 2020 surveys by Ateş and Müller and colleagues, and the former in a 2007 U.K. study [12,13,17]. However, “*Work-to-rule*” seemed to be a greater issue in our sample than in the wards questioned by Ateş and Müller and colleagues [12,13]. “*Increased inter-professional tensions*” occurred less frequently on the NPC ward than on both our neuro ward and the wards surveyed by Ateş and colleagues [13], which could be attributed to the high relevance given to multiprofessional work on our NPC ward. Our results again illustrate the relevance of communication in dealing with dying/death, especially given the background of the multiprofessionality of (neuro)palliative care teams. The factors explaining the differences should be elucidated with a qualitative survey.

All surveyed physical and mental distress symptoms occurred (very) often in over one-third of the participants within the last four weeks, and there were no differences between the two wards. The most common symptoms, “*Feeling rushed*” and “*Feeling completely exhausted after work*”, were consistent with previous results of Ateş and colleagues [13]. However, the staff of the palliative care wards surveyed by Ateş et al., tended to show fewer symptoms of physical and mental distress overall and were considerably more likely to “*Feel calm and balanced*” and “*Full of energy*” [13].

Nearly half of the respondents on both wards regarded every surveyed resource as (very) important, which supports the relevance of the resources queried. Again, the most important resources barely differed, neither between the NPC ward and the neuro ward nor between the NPC ward and the palliative care wards surveyed by Ateş et al. [13]. These resources must be strengthened. “*Team activities outside working hours*” were only considered as (very) important by about half of the respondents, but team activities could be included in working hours, as “*The team*” itself was one of the most important resources. They could, for example, be combined with “*Rituals*”, also a relevant resource indicated. Possible scenarios would be regular joint breakfasts or a joint farewell to deceased patients. “*Private life*” should be empowered; no permanent accessibility should be required, and precise substitution schedules for absences should exist. “*Experiencing nature*” could be supported by outdoor team activities or cheaper and easier access to public transport. A green area of retreat could also be created for the staff, such as a balcony or terrace. Interestingly, on the NPC ward, “*Supervision*” was rated as (very) important by only half of the respondents, while on the palliative care wards in Ateş et al., more than 90% felt this way [13]. This discrepancy should be further investigated, as “*Supervision*” was repeatedly reported to be an essential source for personal and team well-being [18,19].

There were no significant differences in the estimated consistency of the respective team. However, Ateş et al., found that the team tended to be considered slightly more sustainable [13]. The development of working conditions was considered more positively by the teams on the NPC ward and the palliative care wards described by Ateş et al., than on the neuro ward [13]. On both the NPC ward and neuro ward, about one-quarter of respondents indicated that their palliative attitudes had changed since starting work in the respective team, many of them in a positive way. Whereas participants of the neuro ward rather reported an improvement in expertise and awareness, participants of the NPC ward described an improvement in soft skills. Interestingly, interprofessional collaboration was perceived to have improved on the NPC ward, while on the neuro ward, it was reported as worsened. Again, communication seems to play a pivotal role. In contrast, in the study by Ateş et al., the responses were balanced between changed and unchanged palliative care attitudes. However, the change was perceived as deteriorated by 40%, while 16% of them mentioned organizational or institutional conditions that could not precisely be subsumed under a palliative care attitude [13].

### Limitations

Overall, the results must be interpreted with caution. Due to the study’s small sample size and monocentric structure, there is relevant potential for type 2 errors, and there were unmeasured possible confounders (e.g., sociodemographic attributes). In addition, statistically significant results must be interpreted in consideration of multiple testing; we included the Bonferroni-adapted significance levels to adjust for type 1 errors.

However, in the absence of other neuropalliative care units in German-speaking countries, there was no possibility of increasing the sample size, and, as our sample was small, anonymization could not have been guaranteed when collecting more sociodemographic data. The questionnaire only surveyed the subjective perception, which does not depict the objective situation. However, subjective perception can relevantly influence and predict physical and mental health [20].

Since the surveyed NPC ward was operated for one year only and compared to established wards, the results may be reevaluated in another survey after a longer period of operation. When comparing the results to the studies by Müller and Ateş and colleagues, one has to consider that they also surveyed other professional groups and volunteers and that three-quarters of respondents worked part-time [13]. However, the overall comparability is high, not least because we used the same questionnaire and surveyed in the same country with similar regulations.

## 5. Conclusions

The NPC team was not more burdened than the team of the neurological ward, including the stroke unit. Although work was generally considered meaningful and often satisfying, the overall perception of burden was high in both groups. Relevant burdening factors should be identified on a regular basis to be able to address them before relevant overload occurs. The circumstances of death seem to be more relevant than the number of deaths alone. Similar resources were identified that need to be empowered. Communication is simultaneously a relevant burdening factor and a potential resource. Measures should be taken to strengthen communication skills and encourage effective and appreciative communication within a multiprofessional team and with patients and their relatives.

## Figures and Tables

**Figure 1 brainsci-12-01697-f001:**
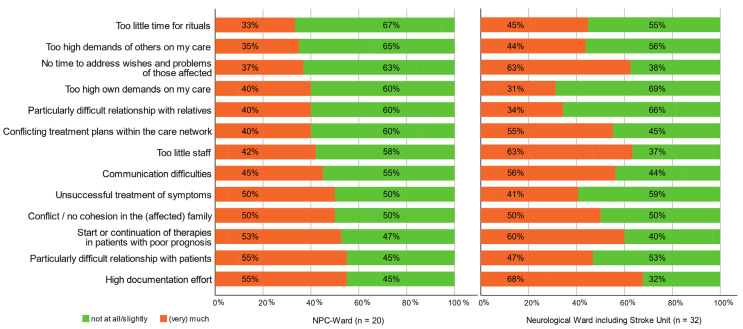
Factors perceived as especially burdening (= over one-third of the team answered with much or very much burdening) on the NPC ward and the neurological ward, including the stroke unit. The importance of burden factors was dichotomized as “not at all/slightly” and “(very) much”.

**Table 1 brainsci-12-01697-t001:** Sample description of the NPC ward and the neuro ward with relative and absolute numbers. Mean and standard deviation (±SD) given. * Statistically significant differences (*p* < 0.05).

	NPC Ward	Neuro Ward	Mann–Whitney U-Test
Response rate	87% (*n* = 20)	78% (*n* = 32)	n.s.
Mean work experience with SD	17.7 years (±14.5)	11.1 years (±10.3)	n.s.
Experience in palliative care	4.7 years (±9.6)	0.9 years (±2.2)	*p* = 0.019 *
Overtime per week	4.2 h (±3.9)	6.3 h (±8.7)	n.s.
			Fisher’s exact test
Living situation	Alone: 35% (*n* = 7)	Alone: 34% (*n* = 11)	n.s.
	Together: 65% (*n* = 13)	Together: 66% (*n* = 21)	
Working time	Full-time: 75% (*n* = 15)	Full-time: 91% (*n* = 29)	n.s.
	Part-time: 25% (*n* = 5)	Part-time: 9% (*n* = 3)	
Care dependent relatives	20% (*n* = 4)	31% (*n* = 10)	n.s.
Profession	Nursing staff 75% (*n* = 15)	Nursing staff 78% (*n* = 25)	n.s.
	Physicians 25% (*n* = 5)	Physicians 22% (*n* = 7)	

**Table 2 brainsci-12-01697-t002:** Perceived frequency of institutional and interpersonal burdening events on the NPC ward and the neuro ward in relative numbers. The occurrence of burdening events is summarized as “never”, “rarely”, “sometimes”, “often”, and “always” to “never/rarely”, “sometimes”, and “often/always”.

	NPC Ward			Neuro Ward			Fisher’s Exact Test
	*Never/* *Rarely*	*Sometimes*	*Often/* *Always*	*Never/* *Rarely*	*Sometimes*	*Often/* *Always*	
Sense of purpose	0%	20%	80%	6%	0%	94%	n.s.
Job satisfaction	5%	35%	60%	9%	34%	57%	n.s.
Excessive responsibility	70%	15%	15%	35%	44%	22%	*p* = 0.05
Wish for more responsibilities	40%	50%	10%	35%	41%	25%	n.s.
Unmanageable workload	50%	20%	30%	47%	13%	40%	n.s.
Interprofessional work meets expectations	10%	30%	60%	13%	50%	37%	n.s.
Failing to partake in educational opportunities due to lack of resources	60%	25%	15%	44%	22%	34%	n.s.

**Table 3 brainsci-12-01697-t003:** Personal resources when dealing with death on the NPC ward and the neuro ward in relative numbers. The importance of resources was dichotomized in “not at all/less important” and “(very) important.” No statistically significant differences (*p* < 0.05) were found.

	NPC Ward		Neuro Ward		Fisher’s Exact Test
	*Not at All/Less Important*	*(Very)* *Important*	*Not at All/Less Important*	*(Very)* *Important*	
Supervision	50%	50%	44%	56%	n.s.
Rituals	35%	65%	56%	44%	n.s.
The team	0%	100%	16%	84%	n.s.
Family	15%	85%	19%	81%	n.s.
Private life	10%	90%	22%	78%	n.s.
Sports	45%	55%	31%	69%	n.s.
Distraction	35%	65%	37%	63%	n.s.
Humor	15%	85%	12%	88%	n.s.
Worldview/Faith	50%	50%	53%	47%	n.s.
Empathy of others	45%	55%	47%	53%	n.s.
Experiencing nature/walks/hikes	25%	75%	37%	63%	n.s.
Meeting friends	20%	80%	19%	81%	n.s.
Offers for self-care	55%	45%	47%	53%	n.s.
Team activities, e.g., excursions, gatherings, team sports	55%	45%	47%	53%	n.s.

**Table 4 brainsci-12-01697-t004:** Changes in palliative care attitude (**A**), expected team sustainability (**B**), and changes in working conditions (**C**) on the NPC ward and neuro ward with relative and absolute numbers. The changes in working conditions were combined to “(strongly) improved” and “(strongly) worsened”. * Statistically significant differences (*p* < 0.05).

	NPC Ward	Neuro Ward	Fisher’s Exact Test
(A) Changes in palliative care attitude			
Changed	25% (*n* = 5)	22% (*n* = 7)	n.s.
Unchanged	55% (*n* = 11)	66% (*n* = 21)	
Indecisive	20% (*n* = 4)	13% (*n* = 4)	
(B) Expected team sustainability			
Few weeks	0% (*n* = 0)	3% (*n* = 1)	
Few months	15% (*n* = 3)	25% (*n* = 8)	n.s.
Few years	35% (*n* = 7)	44% (*n* = 14)	
Many years	50% (*n* = 10)	28% (*n* = 9)	
(C) Changes in working conditions since employment			
(Strongly) improved	45% (*n* = 9)	19% (*n* = 6)	*p* = 0.033 *
Unchanged	40% (*n* = 8)	56% (*n* = 18)	
(Strongly) worsened	15% (*n* = 3)	25% (*n* = 8)	

## Data Availability

Due to privacy issues, the data presented in this study are not available.

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
