# Peer review of "Burdens and Resources of Staff of a Specialized Ward for Neuropalliative Care: A Cross-Sectional Survey"

_brainsci, 2022, doi:10.3390/brainsci12121697_

Round 1

Reviewer 1 Report

Thank you for the opportunity to review this manuscript.

The authors underscore the increasing role of palliative care in health care, and specifically the importance of burdens upon and supports for palliative care staff in relation to both staff wellbeing and sustainable quality of patient care at the end of life. They make a compelling case that health professionals providing neuro-palliative care are likely to encounter distinctive challenges.

The manuscript provides data on original research that is of potential importance to occupational health psychology and health service quality.

The study has substantial merits but I consider that substantial revision is required in order for it to be publishable.

 1. Framing of introduction and justification for the study

2. Consistency of terminology

3. Statistical analysis and interpretation of findings

4. Consideration of limitations

Introduction

The overall framing of the introduction in both the Abstract (lines 8-9) and main text (lines 32-35), suggests that the research hypothesis will address challenges of providing palliative care that are distinctive among patients with life-limiting neurological disorders: "limitations in mobility, cognition … communication [and behavior]". Consequently, I was surprised to discover that the comparison undertaken in the study is actually between NPC staff caring for different types of neurology patients (rather than NPC staff versus other palliative care professionals). Can the authors justify the claim central to the group comparison that NPC involves care of patients who are "severely ill … compared to [patients cared for by] the staff of other neurological wards including a stroke unit"? [lines 61-62] Even if the NPC patients have conditions ultimately more life-limiting on average than non-NPC patients, inpatient status implies that patients are likely to be very unwell, even if only temporarily so. For example, a patient with acute stroke, Guillain-Barre syndrome or exacerbation of multiple sclerosis requiring hospital admission may be "severely ill" while admitted even if their condition has good prospects for future recovery.

Methods

The overall content and categorization of items in the adapted Ateş questionnaire is difficult for a reader to understand:  In Methods (lines 72-75), it is stated briefly that items target "stress factors, stress symptoms, supportive factors, and physical and mental well-being … palliative care attitude, change of working conditions and team sustainability, team interaction and stress perception were surveyed." However, the Results are not presented according to this categorization of items, or at least not with the same terminology. There needs to be consistent use of terms and categories across Methods and Results need to correspond exactly.

The categorization of items is particularly important given the decision "compare main tendencies … all answers of the respective category were pooled and compared" [lines 90-91] It is conventional methodology for pooling of combined questionnaire items to validate the basis for this combination of individual item responses, e.g., with exploratory factor analysis. Can the authors justify use of pooled items without such validation?

The statistical comparisons are reportedly all based on Mann-Whitney-U test. However, many of these comparisons appear to be tests of 2x2 proportions for which a Fisher's exact test would be more appropriate. Can the authors justify their appraoch to statistical testing?

"Group differences 91 with p < 0.05 were considered as statistically significant." [line 91-92] Given the very large number of statistical tests presented throughout the Results, there needs to be either an adjustment of the p-value threshold for multiple tests (most traditionally, e.g., the Bonferroni method), or at the very least a caveat in the 'Limitations' section on interpretation of statistically significant findings in consideration of the multiple tests.

Results

In comparing group characteristics [Table 1; following line 110], the magnitude of differences between groups in the study sample is more important than statistical inferences (of differences in relation to theoretical populations from which the sample is drawn). For example, the substantial difference in mean work experience (17.7 versus 11.1) may well be an important confounder of responses between the two groups, even though this difference is nominally "not significant" (in terms of statistical inference) because of the small sample size. It is not correct to assert that "mean work experience did not differ [line 99]".

While it is understandable that within-institution anonymity considerations precluded collection of information on age and sex [lines 105-107], these demographic attributes are potentially important confounders of the between-group comparisons.

Discussion

It is conventional to open a Discussion section with a brief contextualising summary of study findings, rather than with an extension/repetition of the background/introductory literature required to justify the study, as is currently presented in the MS. This is disorientating for a reader and needs to be reorganized, e.g., a sentence similar to the one currently in lines 232-234 ought to open the Discussion section.

Considering the small sample size and consequent imprecision of estimates, along with distinctive sample attributes 'unknowns' (e.g., basic demographic characteristics), I am not convinced that comparisons with national averages [lines 243-247] are meaningful

Fundamental to occupational health psychology are the concepts of demand-control imbalance (job strain') and effort-reward imbalance, for which adverse health outcomes have been demonstrated in multiple large cohort studies. It would be helpful for the study findings to be contextualised briefly in regard to occupational health psychology more broadly.

Limitations [lines 338 ff]: The findings of the study should be considered very tentative. In relation to the statistical significance and overall validity of results, there is considerable potential for Type 1 error (especially due to multiple testing with uncorrected p-value threshold), hasty generalisation from unvalidated pooling of items, Type 2 error from small sample size, and unmeasured confounding (e.g., from differences in the demographic attributes and work experience of staff in the two groups).

Author Response

Thank you for the substantial and detailed review, which gaves us the opportunity to improve our paper.

Reviewer 2 Report

Dear authors, 

Thank you for letting me review your excellent work here, congratulations. 

I think is necessary to expose these data, although the results seem not significant at the first moment, a qualitative analysis could improve and enhance the conclusions.

I see some minor issues:

The abstract and introduction are son clear, complete, and sufficient to read. 

The method: Line 71: Please explain An adjusted version of the questionnaire by Ateş et al [13] was used. If you don't use a neat version of the questionnaire, you need to analyze its consistency, and the alpha Cronbach test must be performed. This issue is important and relevant. 

Line 86: Please cite IBM SPSS as IBM Corp. Released 2016. IBM SPSS Statistics for Windows, Version 24.0. Armonk, NY: IBM Corp

I usually see how to cite here: https://www.ibm.com/support/pages/how-cite-ibm-spss-statistics-or-earlier-versions-spss 

I think the discussion is right, and the authors point to the differences (small differences) between aspects of palliative care in some kind of patients.

Reviewer 3 Report

Interesting and easy-to-read study (although it has a lot of detailed information).

The objective was presented clearly. The results were presented to respond to it and were well structured.

The discussion is carried out considering the existing evidence.

Although it is a study in a particular context and carried out in a unit with a relatively short period of existence, the truth is that it is in this period of training of the teams that they potentially go through crises due to the possible lack of identification with the philosophy of palliative care. This is to tell that it makes sense to carry out this study, although its period of existence is a limitation, but the authors identify this.

Adjusted conclusion, with a practical sense of research gains.

I suggest adding a reference, which, because it is a systematic review and because it is specifically about comparing PCs with other contexts, it makes sense that it can be used:

Burnout in palliative care settings compared to other settings: a systematic review

doi: 10.1097/NJH.0000000000000370

Round 2

Reviewer 1 Report

I thank the authors for their assiduous efforts in responding point-by-point to my original review. I consider that the manuscript is nearly ready and does not require another round of rebuttal/revision and review.

One minor point: note that, in relation to study limitations, the potential for Type 2 error and confounding are different issues. The relevant sentence [lines 366-368] ought to be reorganized along the lines of: "Due to the small sample size ... there is potential for Type 2 error, and there were unmeasured possible confounders (e.g., sociodemographic attributes)."

The English language standard of the manuscript remains somewhat problematic and will require thorough attention in final copyediting. As one of many potential examples of awkward wording, the authors have manifestly struggled (lines 61-62) to find the correct past tense of the verb 'arise', replacing 'arised' with 'aroused', where 'arose' would be more appropriate.

Author Response

Thank you for your comments.

One minor point: note that, in relation to study limitations, the potential for Type 2 error and confounding are different issues. The relevant sentence [lines 366-368] ought to be reorganized along the lines of: "Due to the small sample size ... there is potential for Type 2 error, and there were unmeasured possible confounders (e.g., sociodemographic attributes)."

We revised the sentence as suggested (line 366-368).

The English language standard of the manuscript remains somewhat problematic and will require thorough attention in final copyediting. As one of many potential examples of awkward wording, the authors have manifestly struggled (lines 61-62) to find the correct past tense of the verb 'arise', replacing 'arised' with 'aroused', where 'arose' would be more appropriate.

We would like to apologize for the grammatical error(s).